

# Foot darkening with age in *Spheniscus* penguins: applications and functions

Ginger A. Rebstock,  K. Pearl Wellington and  P. Dee Boersma

Biology, University of Washington, Seattle, WA, United States of America
Center for Ecosystem Sentinels, Seattle, WA, United States of America

## ABSTRACT

The proportions of individuals in various age classes in a population of wild animals affect population trends, behaviors, learning, and social structures. Knowledge of age structure is needed for effective conservation and management of populations. However, it is not always possible to determine the age or age class of individual animals, and hence the age structure of the population. Penguins, like most birds, cannot be aged once they molt into adult plumage. *Spheniscus* penguins attain adult plumage at 6 to 24 months of age, and individuals can live more than 30 years. We studied foot darkening in the four species of *Spheniscus* penguins to determine if age class can be determined from foot color. We compared how foot color changes with age among the four species to investigate potential functions of the darkening. We found that *Spheniscus* penguins have pale feet at hatching and the feet become darker with age throughout the lives of individuals. We showed that we can accurately predict the age structure of a colony of Magellanic penguins *Spheniscus magellanicus*, but not the ages of individual penguins, based on a sample of foot colors. The timing of foot darkening within species was consistent with foot color functioning in protection from UV radiation, and not with foot color functioning in thermoregulation. The species that breeds at the lowest latitudes and experiences the highest UV radiation (Galápagos penguins *Spheniscus mendiculus*) had feet that darkened at the earliest ages, and the species that breed at higher latitudes and experience less intense insolation (African *S. demersus* and Magellanic penguins) had feet that darkened latest. Humboldt penguins *S. humboldti* breed mostly at low latitudes and foot darkening was intermediate between Galápagos and Magellanic penguins. We also found that males' feet darken somewhat earlier than females' feet, likely because females spend more time in their nests (burrows or under vegetation) than males and have less sun exposure. We found that feet darkened in an individual over years, but not within a breeding season. The color change is a life-long process, likely an evolutionary adaptation within species, not a seasonal, temporary response to UV radiation. We propose foot darkening as a way to assess age structure in *Spheniscus* penguins. Foot color in a colony of Magellanic penguins can provide a rapid, noninvasive method to estimate the age structure of the colony.

Corresponding authors
Ginger A. Rebstock,
gar@u.washington.edu
K. Pearl Wellington, pearl42@uw.edu

## INTRODUCTION

Age structure of a population, that is, the proportion of the population in each age class, affects demographics and population trends (*Coulson et al., 2001*; *Bjørnstad, Nisbet & Fromentin, 2004*). Population models that do not consider age may overestimate or underestimate the risk of extinction (*Roach & Carey, 2014*). The proportion of older individuals in a population can have negative or positive effects on population growth, as reproductive performance and survival depend on age in many plant and animal populations (*Jones et al., 2014*; *Roach & Carey, 2014*). For example, older individuals may have lower survival, reproductive output, or foraging efficiency than younger individuals because of senescence (*Monaghan et al., 2008*). Alternatively, older individuals may be more fecund than younger individuals, especially in species that grow larger in size throughout life (*Birkeland & Dayton, 2005*). In addition, older individuals may increase population persistence because younger individuals learn migration and other strategies from more experienced individuals (*Mueller et al., 2013*; *MacCall et al., 2018*).

Effective conservation of species and management of exploited populations rely on knowledge of age structure. Changes in population age structure can be an early-warning signal of delayed responses to environmental changes in long-lived species (*Coulson et al., 2001*; *Holmes & York, 2003*), allowing conservation actions to be taken before a population decline is obvious. In addition, fisheries that remove specific age classes from a population affect productivity and resilience of the population (*Griffiths et al., 2024*). Taking age into account is also necessary for management of trophy hunting, as removal by hunters of males at their prime ages affects social structure and behaviors such as infanticide, with effects on the population (*Roach & Carey, 2014*).

Many methods of aging individuals have been proposed for various animal and plant species, depending on the anatomy, physiology, and natural history of the species. Some taxa have growth increments or rings in hard parts that can be counted to determine age, such as teeth in mammals, otoliths in fish, and wood in trees (*Evans et al., 2021*). However, precise aging with growth increments often requires sacrificing the individual. Moreover, many taxa lack structures with growth increments that are useful for aging. Mark-recapture models are used to determine age structure in wild populations (*Sidhu, Catchpole & Dann, 2007*; *Zajitschek et al., 2009*). However, it is not always possible to capture and mark or to recapture individuals, and for long-lived species, years or decades are required for marked young to mature and reach older age classes. Molecular techniques are increasingly being used to age wild animals (*Juola et al., 2006*; *De Paoli-Iseppi et al., 2018*). However, determining population age structure by molecular methods requires capturing large numbers of individuals and taking blood or skin samples. Moreover, these methods do not work on all species (*Juola et al., 2006*; *Cerchiara et al., 2017*). Approximate ages or age classes can be determined by noninvasive observations or analysis of photographs in some species, using, for example, the amount of scarring in mammalian predators (*Miller et al., 2016*), and the nose color in some large mammals (*Van Horn et al., 2015*; *Miller et al., 2016*).

Birds lack structures with growth increments that can be used to determine exact age, but age class can be determined in some cases. For example, most birds have a juvenile plumage, which is kept for weeks to years (*Moreno & Soler, 2011*). Large, long-lived birds often have several age-specific immature plumages (*Prince et al., 1997*). In addition, iris color changes with age in many bird species, either between immature and adult stages (*Polakowski et al., 2020*), or throughout life in a few species (*Scholten, 1999*; *Bortolotti, Smits & Bird, 2003*). Adult age class can be determined by specific structures in a few species, such as Atlantic puffins *Fratercula arctica*, which can be assigned an age class based on grooves in the bill (*Harris, 2014*). In addition, bare-part color sometimes changes with age in birds (*Nicolaus et al., 2007*).

Bare-part colors in birds, including extinct taxa (*Roy et al., 2020*), have various functions, including signaling, thermoregulation, and protection from ultraviolet (UV) radiation or abrasion (*Savalli, 1995*; *Galván & Solano, 2016*). Colors used in signaling, often bright colors (*Galván & Solano, 2016*; *Iverson & Karubian, 2017*), depend more on diet, condition, or breeding status than on age in adults (*Laucht & Dale, 2012*) and are often sex specific (*Scholten, 1999*; *Lewis et al., 2020*) or seasonal (*Scholten, 1999*). Pigments that provide protection or thermoregulation, often dark, melanin-based colors (*Galván & Solano, 2016*; *Iverson & Karubian, 2017*), are better suited to aging individuals than colors used in signaling, as protective or thermoregulatory pigments may change with age or age class rather than breeding status or body condition.

We studied the relationship between foot color and age in the four species of "banded" penguins, African (*Spheniscus demersus*), Galápagos (*S. mendiculus*), Humboldt (*S. humboldti*), and Magellanic (*S. magellanicus*). *Spheniscus* penguins are referred to as "banded" because adults have black and white bands (stripes or curves) on their faces, chests, and flanks. The term does not refer to bands or rings used to mark individual penguins. Banded penguins cannot be aged based on plumage once they molt into adult plumage (6 months—2 years of age; Table 1). Given the longevity of banded penguins (about 20–30 years; Table 1), and the brief period when they can be aged by plumage, we sought a method of determining the age class of adults.

We compared the timing of foot darkening among the four *Spheniscus* species to test two hypotheses about the function of foot darkening: UV protection and thermoregulation. We assumed that dark colors are unlikely to function in signaling, as most signaling colors are bright (*Galván & Solano, 2016*), and that the tops of the feet need more protection from UV radiation than from abrasion. We found *Spheniscus* penguin chicks' feet are pale at hatching and darken with age. If dark pigments in feet serve as protection from UV radiation (*Fox, 1962*; *Gómez et al., 2018*; *Nicolaï et al., 2020*), feet in species that evolutionarily experienced the most insolation should darken at earlier ages ($H_1$). Using latitude as a proxy for insolation (*Meador et al., 2009*), we predicted the order of foot darkening: Galápagos >Humboldt >African >Magellanic penguins. If pigments function in thermoregulation (*Burtt Jr, 1986*; *Savalli, 1995*) with dark feet absorbing more heat than pale feet, we predicted that feet in species that swim in colder waters (*Palacios, 2004*; *Belkin, 2009*) should darken at earlier ages ($H_2$): Magellanic >Humboldt >African >Galápagos penguins. We used water

**Table 1  Ages at molt into adult plumage and first breeding in *Spheniscus* penguins, and maximum known age.**

| Species | Age at molt into adult plumage (years) | Median age at first breeding (years) | Maximum known age (years) | References |
|---|---|---|---|---|
| African *S. demersus* (Linnaeus 1758) | 1–2[1] | 6[2] | 27[3] | [1](*Kemper & Roux, 2005*) [2](*Whittington et al., 2005*) [3](*Whittington, Dyer & Klages, 2000*) |
| Galápagos *S. mendiculus* Sundevall 1871 | 0.5[1] | Unknown | 17.7[2] | [1](*Boersma, 1977*; *Boersma et al., 2013b*) [2](*Jiménez-Uzcátegui & Vargas, 2019*) |
| Humboldt *S. humboldti* Meyen 1834 | 0.5–1.5[1] | 6[2] | ~30[3] | [1]Arianna Basto, Punta San Juan Program, pers. comm., 2024 and zoo sample [2] (*De la Puente et al., 2013*) [3](*Johnson, 2022*) |
| Magellanic *S. magellanicus* (Forster 1781) | 1[1] | 6-8[1] | >30[1] | [1](*Boersma et al., 2013a*) |

temperature rather than air temperature, as bodies lose heat faster in water than in air (*Stahel & Nicol, 1982*) and penguins spend much of their time foraging in the ocean.

Males spend more time at breeding colonies (*Boersma, Stokes & Yorio, 1990*) and more time outside their nests in the colonies, calling and fighting (*Renison, Boersma & Martella, 2003*; *Gownaris, García Borboroglu & Boersma, 2020*), than females. We predicted that males' feet would darken earlier than females' feet because of the greater UV exposure in males than in females (H$_3$). We compared females and males within each species as the difference between the sexes may depend on the species.

We also tested seasonality of foot color in adult Magellanic penguins, with a large sample throughout breeding seasons. If foot darkening is a temporary response to UV exposure, we would expect feet to be lighter at the beginning of each breeding season, and darker later in the season (*Nicolaï et al., 2020*). Conversely, if foot darkening is an evolutionary adaptation to UV exposure in the breeding habitats, pigments should accumulate continuously (H$_4$), and there should be no seasonal cycle.

Finally, we tested whether we can estimate the age structure of a population of Magellanic penguins based on foot color in a sample of individuals. This would be a rapid, noninvasive method of determining age structure in a breeding colony. Among the four *Spheniscus* species, we only had a large enough sample size of known-age penguins to test this for Magellanic penguins.

## MATERIALS & METHODS
### Study populations
We used a captive population of African penguins *Spheniscus demersus* at San Diego Zoo, San Diego, California, USA. We coded foot color (see below) from photographs taken in 2022 and 2023. All African penguins were in adult plumage, from 1 to 29 years of age. In the wild, African penguins breed in Namibia and South Africa, between about 24.6°S and 34.6°S.

We used a marked, known-age population of wild Galápagos penguins *S. mendiculus* at the Galápagos Islands (Ecuador) of Bartolomé, Fernandina, Floreana, Isabela, and Santiago. Penguins were marked with web tags (*Boersma & Rebstock, 2010*) when they were in juvenile plumage. We coded foot color in the field between 2015 and 2022. The sample included adults from 1 to 10 years of age, penguins in juvenile plumage, and chicks (nestlings). Galápagos penguins breed in the equatorial Galápagos Islands, about 0.2°N–1.3°S.

We used a captive Humboldt penguin *S. humboldti* population at the Woodland Park Zoo, Seattle, Washington, USA, and coded foot color from photographs taken between 2015 and 2019. This sample also included adults (from 1 to 26 years of age), juveniles, and chicks. Humboldt penguins breed in Chile and Peru, between 5.2°S and 42.2°S, although most of the population breeds north of about 33.3°S.

We used a marked, known-age population of wild Magellanic penguins *S. magellanicus* at Punta Tombo, Chubut Province, Argentina, studied since 1982 (*Boersma, Stokes & Yorio, 1990*). Penguins were marked as nestlings or in juvenile plumage with stainless-steel flipper bands or web tags (*Boersma & Rebstock, 2010*). We coded foot color in the field between 2014 and 2023. This sample included adults from 2 to 31 years of age, juveniles, and chicks. Magellanic penguins breed in Argentina, Chile, and the Falkland Islands (Islas Malvinas), between about 32.5°S and 56°S, although colonies north of about 45°S in Argentina are relatively recent (*Boersma, Stokes & Yorio, 1990*).

All field work was approved by the Institutional Animal Care and Use Committee of the University of Washington (Protocol #2213-02; IACUC approval #TR202100000016). Galápagos National Park approved field work in the Galápagos. The Offices of Fauna and Flora and of Tourism of Chubut Province, Argentina, approved field work in Argentina. John Samaras and Celine Pardo at the Woodland Park Zoo, and Debbie Denton and Megan Owen at the San Diego Zoo provided access to penguins and/or foot photographs, and information on their penguins.

We combined ages into five age classes (Tables 2 and 3) because there were too few older penguins, except in Magellanic penguins, to model age in years. Chick and juvenile age classes were based on plumage. Chicks were covered in down and still associated with their nests in the wild populations. Juveniles had waterproof feathers and lacked the distinct bands on the face, chest, and flanks that adults have (*García Borboroglu & Boersma, 2013*). The adult age classes were based on median age of first reproduction (Table 1) and age at onset of senescence. We combined pre-breeders in adult plumage and young breeders into the pre-breeder category. Age at first breeding varies within species (*Whittington et al., 2005*; *Boersma et al., 2013a*; *De la Puente et al., 2013*), and we preferred to include some young breeders with pre-breeders rather than to include many pre-breeders with middle-age breeders. In addition, pre-breeders spend less time at colonies than breeders, and have low detection rates (*Gownaris & Boersma, 2019*). The division between middle-age and elder breeders is based on the age at which survival starts to decline linearly with age (*Gownaris & Boersma, 2019*) and females lay smaller eggs (*Cerchiara et al., 2017*) in Magellanic penguins. We assumed African and Humboldt penguins are similar to Magellanic penguins in the onset of senescence, as their life expectancies are similar. Age at first breeding and onset

**Table 2  Ages of *Spheniscus* penguins in the five age classes we used in the analyses.** Chicks and juveniles are based on plumage. Breeders are based on median age at first breeding and onset of senescence (Table 1).

| Species | Chicks (years) | Juveniles (years) | Pre-breeders/young breeders (years) | Middle-age (years) | Elder (years) |
|---|---|---|---|---|---|
| African | 0 | 1–2 | 2 or 3–8 | 9–18 | 19+ |
| Galápagos | 0 | 0.5 | 1–3 | 4–10 | 11+ |
| Humboldt | 0 | 0.5–1.5 | 1 or 2–8 | 9–18 | 19+ |
| Magellanic | 0 | 1 | 2–8 | 9–18 | 19+ |

**Table 3  Number of known-age *Spheniscus* penguins by species, age class, and sex used in our analyses.** We used multiple observations for individuals that had more than one foot-color code within a year (4% of adults, < 1% of juveniles), or observations that were at least 1 year apart.

| Species | African | | Galápagos | | Humboldt | | Magellanic | |
|---|---|---|---|---|---|---|---|---|
| | Female | Male | Female | Male | Female | Male | Female | Male |
| Total observations | 27 | 23 | 78 | 76 | 51 | 75 | 151 | 665 |
| Chicks | 0 | 0 | 1 | 0 | 6 | 10 | 2 | 12 |
| Juveniles | 0 | 0 | 62 | 68 | 15 | 19 | 65 | 34 |
| Pre-breeders | 18 | 18 | 10 | 7 | 27 | 38 | 27 | 135 |
| Middle-age | 7 | 2 | 5 | 1 | 1 | 5 | 40 | 353 |
| Elder | 2 | 3 | 0 | 0 | 2 | 3 | 17 | 131 |
| Individuals | 15 | 12 | 66 | 67 | 27 | 37 | 121 | 400 |
| Observations per individual (min-max) | 1–2 | 1–2 | 1–4 | 1–4 | 1–4 | 1–6 | 1–4 | 1–6 |

of senescence are not known in Galápagos penguins (*Boersma et al., 2013b*). It is likely earlier than in the other three species, as Galápagos penguins have shorter lifespans (*Jiménez-Uzcátegui & Vargas, 2019*) and molt into adult plumage younger (*Boersma, 1977*; *Boersma et al., 2013b*).

The captive African and Humboldt penguins were sexed using standard DNA analysis for birds. Woodland Park Zoo and San Diego Zoo sent samples to Animal Genetics/Avian Biotech (Tallahassee, Florida, USA) for sexing. We sexed Galápagos penguins by bill size (*Cappello & Boersma, 2018*) or breeding behavior. We sexed Magellanic penguins by bill size (*Boersma et al., 2013a*; *Rebstock & Boersma, 2023*), cloaca size (*Boersma & Davies, 1987*), or breeding behavior. If we were uncertain of the sex, we excluded the individual.

## Foot color

We noted the color of the tops (dorsal surfaces) of the feet and legs (tarsometatarsus) below the feathering. We defined four categories of foot color: white, mixed, black & white, and mostly black (Fig. 1). White (WHI) includes white, gray, and pinkish feet. White feet may have a few dark spots or be very mottled, but have very little solid black. Mixed (MIX) feet have solid patches of black. Pink is visible on more than just the nail (claw) fold and upper tarsometatarsus. Black & white (B&W) feet have extensive black with black pigment appearing to travel down the tops of the feet. Black & white feet still have some white or

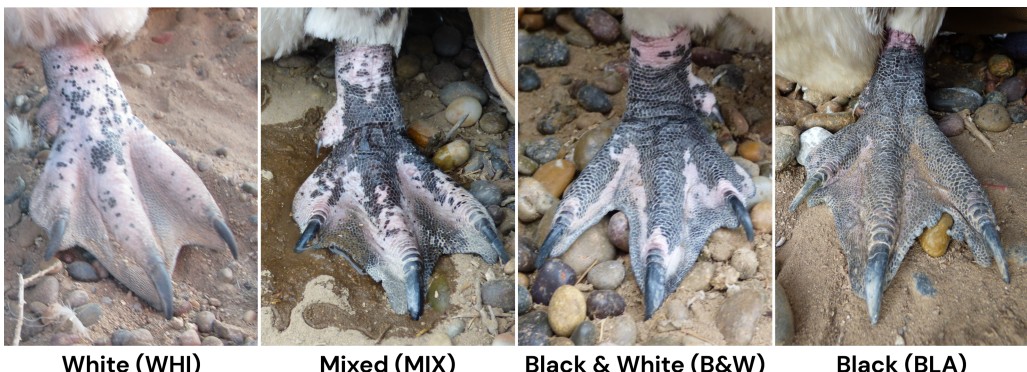

| White (WHI) | Mixed (MIX) | Black & White (B&W) | Black (BLA) |

**Figure 1** **Examples of foot colors in Magellanic penguins.** Photo credit: Dee Boersma.

pink, especially on the nail fold and occasionally the tarsometatarsus. A distinct pinkish border surrounds the toenails. Black (BLA) feet are mostly or all black. There may be a faint pink border in the webbing, but the pink does not extend up the nail fold.

## Timing of foot darkening

To test hypotheses, we compared species and sexes using multinomial logistic regression (mlogit) in Stata IC11.2 (StataCorp, College Station, TX, USA). Foot color, as an unordered factor, was the response variable. Although foot color is ordered, our data did not support the proportional odds ratio assumption, so we used multinomial rather than ordinal regression. Ordinal regression constrains odds ratios between pairs of levels of a variable to be the same. If odds ratios differ among levels of a variable, odds ratios are distorted and ordinal regression is not valid. Multinomial regression returns log-odds ratios between levels of the predictor variables, which are essentially the slopes, or relative rates of change in the response variable. Multinomial regression allows these slopes to vary among levels of the variables. We specified robust standard errors clustered on individual ID (*Long & Freese, 2006*), which adjusts the degrees of freedom for the number of clusters (individual penguins). Predictors were species, age class, and sex.

We calculated predicted probabilities from the coefficients (log-odds ratios) of the variables, using the Stata predict command. We compared the probabilities of having each foot color for each species and age class to test hypotheses $H_1$ (foot pigment functions as sunscreen) and $H_2$ (foot pigment functions in thermoregulation). We compared the probabilities of having each foot color at each age between sexes in each species to test $H_3$ (feet darken at earlier ages in males than in females).

## Seasonal foot darkening in Magellanic penguins

If foot darkening is an immediate, temporary response to exposure to the sun, similar to freckling or tanning in humans, foot color should be lightest at colony arrival and darken within each breeding season. If foot darkening is an evolutionary response to sun exposure in the population, feet should darken with increasing age, but not between the beginning and end of a breeding season.

We tested for seasonal darkening ($H_4$) in adult Magellanic penguins' feet using multinomial regression with foot color as the response. Predictors were age in years, sex, and days since 1 September (the beginning of the breeding season). We specified robust standard errors clustered on individual ID. We used a Wald test (mlogtest, wald command in Stata) to obtain test statistics for each predictor.

**Predicting adult age structure from a sample of foot color**

To determine if we can predict the adult age structure from a sample of foot color, we used a randomization procedure with 1,000 iterations. We had 777 observations of 511 individual known-age adult Magellanic penguins at Punta Tombo, including 74 penguins of unknown sex. We used each penguin once, selecting one observation for each penguin at random in each iteration. Two penguins had 6 observations each, two had five observations, 19 had four observations, 35 had three observations, 121 had two observations, and 332 had one observation. We classified each penguin using the three adult age classes from Table 2. We wrote custom code in R (*R Development Core Team, 2020*) for the procedure, using tidyverse (*Wickham et al., 2019*) and coin (*Hothorn et al., 2006*) packages.

We then selected 2/3 of the sample at random in each iteration to calculate percentages (model sample, $n = 341$) and 1/3 to test the model (test sample, $n = 170$). We calculated the model percent as the percentages of each age class within each foot color (*i.e.,* colors sum to 100%) in the model sample (Fig. 2). We calculated the test counts as the number of penguins in the test sample with each foot color, regardless of age class. We then multiplied the test counts by the model percentages to get the predicted counts and summed the predicted counts in each age class to get the predicted age structure. We compared the predicted age structure with the observed age structure in the test sample (counts in each age class regardless of foot color) with a $\chi^2$ test (Fig. 2). We counted the number of *p*-values from the 1,000 iterations that were less than 0.05.

# RESULTS

**Function of dark foot pigments: sunscreen or thermoregulation**

Feet got darker with age in all species, both within individuals for which we had up to six observations (Fig. 3) and across the populations. The timing of darkening varied among species and between sexes (Table 4; Fig. 4). The timing of foot darkening among species mostly supported our prediction of foot pigment serving as a sunscreen ($H_1$). Galápagos penguins' feet darkened earlier than the other species' feet. Humboldt penguins' feet darkened earlier than African and Magellanic penguins' feet through middle age. Elder Humboldt penguins were less likely to have black feet than elder Magellanic penguins, however. African and Magellanic penguins' feet darkened at similar ages.

There were exceptions to the relationship between age class and foot color in all species, meaning an individual could not be aged with certainty. Among African penguins, one of three old breeders had mixed feet and three of 20 pre-breeders had black feet. Among Galápagos penguins, one of four middle-age breeders had mixed feet and seven of 128 juveniles had black feet. Among Humboldt penguins, one of four elder breeders had mixed

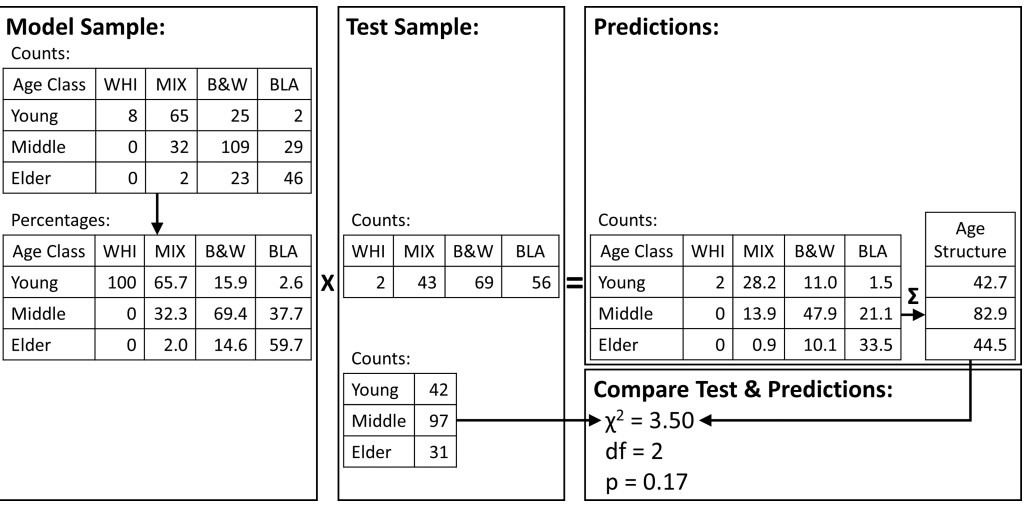

**Figure 2 Randomization procedure used to test predictions of population age structure from a sample of foot colors of adult Magellanic penguins.** Our sample of known-age adults was split into model and test samples at random 1,000 times. For each iteration, we calculated the percentages of each foot color by age class in the model sample (left panel). We counted the foot colors in the test sample (middle panel) and multiplied them by the model-sample percentages. We summed the resulting penguins in each age class (upper right panel) and compared them to the counts in the test sample with a $\chi^2$ test (lower right panel).

**Table 4 Results from multinomial regression of foot color on species, age class, and sex in *Spheniscus* penguins.** Wald $\chi^2$ (15 degrees of freedom) = 4,785, $p < 0.0001$, pseudo R2 = 0.27, $n = 1,146$, robust standard errors adjusted for 745 clusters in penguin ID. All log odds ratios (standard errors in parentheses) are relative to Black & White (the reference level, not shown). Species are relative to African penguins. Sex is relative to females.

| Variable | White | | | Mixed | | | Black | | |
|---|---|---|---|---|---|---|---|---|---|
| | Log odds ratio (se) | z | p | Log odds ratio (se) | z | p | Log odds ratio (se) | z | p |
| Species: Galápagos | 9.7 (0.7) | 13.5 | <0.001 | −2.7 (0.5) | −5.4 | <0.001 | 1.5 (0.8) | 1.9 | 0.06 |
| Species: Humboldt | 11.7 (0.7) | 16.5 | <0.001 | −0.5 (0.5) | −1.0 | 0.32 | −2.7 (1.2) | −2.3 | 0.02 |
| Species: Magellanic | 13.1 (0.7) | 19.2 | <0.001 | 0.004 (0.5) | 0.01 | 0.99 | −0.1 (0.6) | −0.1 | 0.93 |
| Age Class | −4.0 (0.5) | −8.8 | <0.001 | −1.3 (0.2) | −8.2 | <0.001 | 1.6 (0.2) | 10.5 | <0.001 |
| Sex: Male | −0.6 (0.4) | −1.7 | 0.09 | −0.6 (0.2) | −2.6 | 0.009 | −0.3 (0.3) | −1.1 | 0.29 |
| Intercept | −7.8 (1.1) | −7.4 | <0.001 | 3.4 (0.5) | 6.3 | <0.001 | −5.4 (0.9) | −6.1 | <0.001 |

feet and five of 34 juveniles had black & white feet. Among Magellanic penguins, nine of 119 elder breeders had mixed feet and 12 of 99 juveniles had black & white feet.

### Feet darken at earlier ages in males than in females

As predicted, feet generally darkened earlier in males than in females (H3). Darkening timing differences were smaller between the sexes than among the species. Females and males had similar probabilities of having white feet. Adults of both sexes had low or zero probabilities of having white feet. Females were more likely than males to have mixed feet from juvenile through middle age (Fig. 5, Table 4). Females were less likely than males

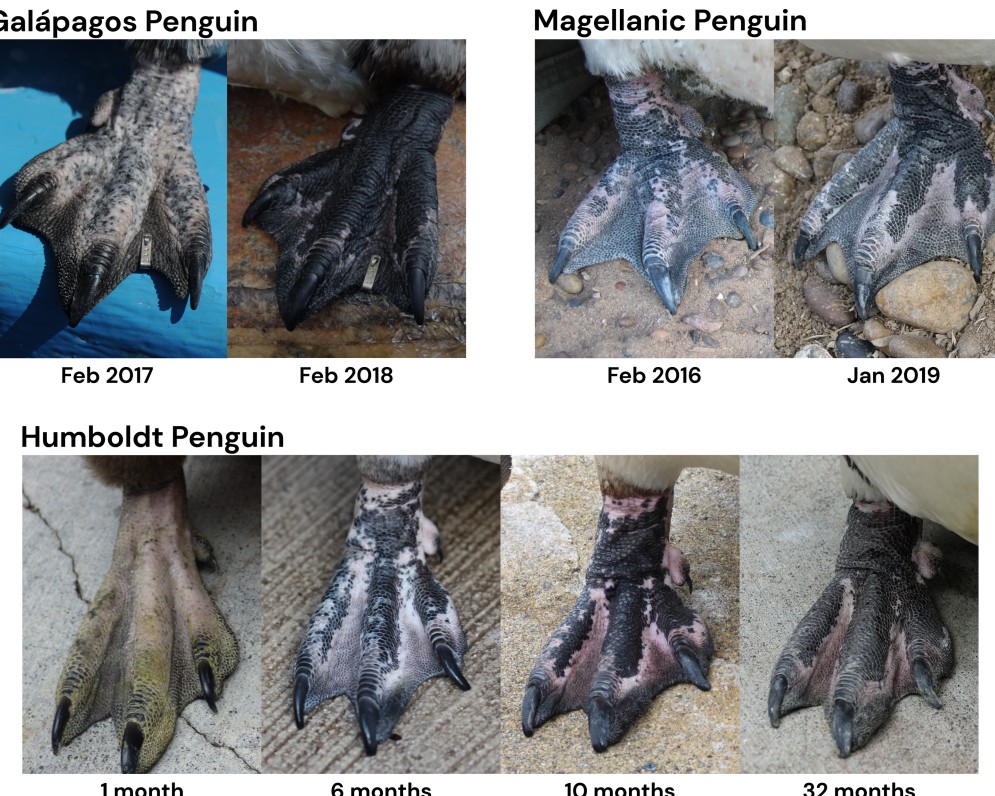

**Figure 3** **Examples of foot color darkening over time in three individuals of *Spheniscus* penguins.**
Galápagos penguin (top left two photos): juvenile (left) and the same individual as a pre-breeder adult. The foot is marked with a web tag. Photo credit: Dee Boersma. Magellanic penguin (top right two photos): juvenile (left) and the same individual as a pre-breeder adult. Photo credit: Dee Boersma. Humboldt penguin (bottom four photos): one individual as a chick (nestling), juvenile, juvenile, and pre-breeder adult (left to right). Photo credit: Pearl Wellington (three left photos) and Dee Boersma (right-most photo).

to have black & white feet from juvenile through old age (Fig. 5). Females and males had similar probabilities of having black feet, with little probability of black feet until middle age, except in Galápagos penguin pre-breeders (Fig. 5).

## Feet did not darken seasonally in Magellanic penguins

In Magellanic penguin adults, feet darkened with age, but not within breeding seasons ($H_4$). Only age (years) was significant ($\chi^2(3) = 189.5$, $p < 0.001$). Foot color was not related to days since 1 September ($\chi^2(3) = 2.6$, $p = 0.47$) or sex ($\chi^2(3) = 5.5$, $p = 0.14$).

## Predicting age structure

Age structure was predicted correctly ($p > 0.05$) in 989 of 1,000 iterations (Fig. 6). *P*-values from the 1,000 iterations ranged from 0.007 to 0.999, with a mean of 0.62. Half of the iterations resulted in *p*-values $> 0.64$.

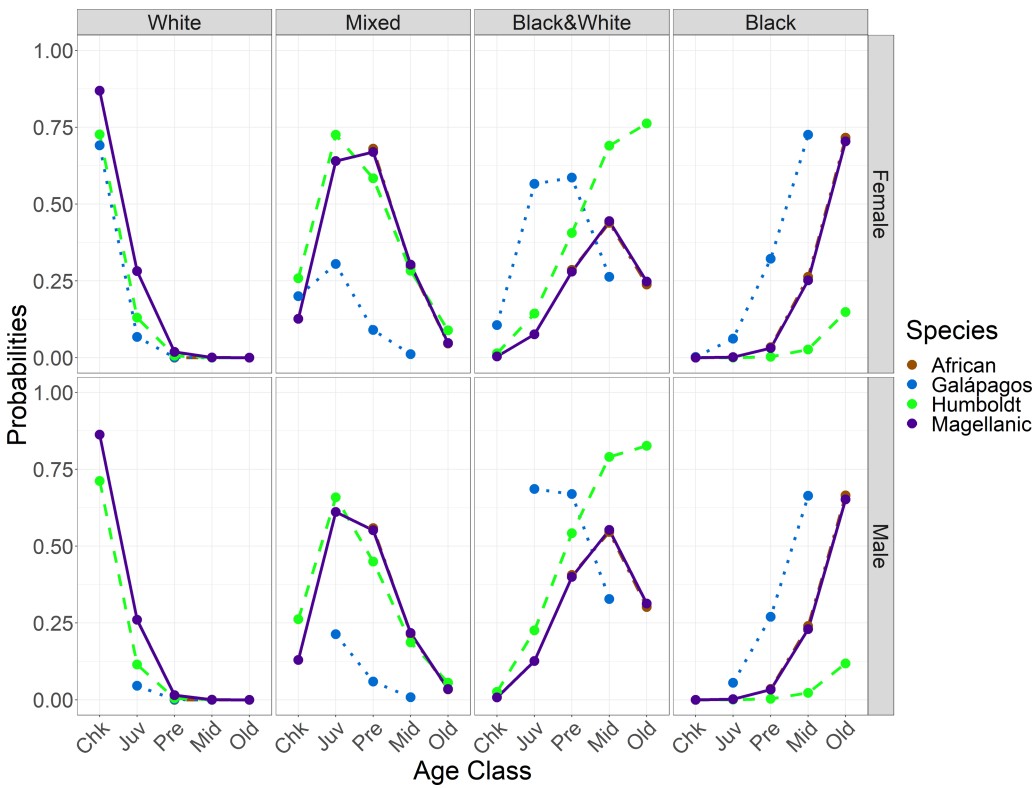

**Figure 4 Probabilities of having each foot color in *Spheniscus* penguins, by age class and species for each sex.** Age classes: Chk, chick; Juv, juvenile; Pre, pre-breeder; Mid, middle age; Old, elder penguin (see Table 2 for age class definitions).

## DISCUSSION

We showed that banded penguin individuals' feet darken throughout life, and the dark pigments likely function to protect tissues from UV damage ($H_1$). We assumed the dark pigments in penguins' feet are melanin (*Nicolaï et al., 2020*), but did not analyze the skin. The UV protection and thermoregulation hypotheses had mostly opposite predictions regarding the timing of foot darkening among banded penguin species, and we found support for UV protection, not thermoregulation, similar to black skin in birds of many taxa (*Nicolaï et al., 2020*). Feet of the species that experienced the highest insolation (Galápagos followed by Humboldt penguins) darkened earliest. African and Magellanic penguins' feet darkened later, and at similar rates to each other. Note that UV radiation increases at high southern latitudes (*Meador et al., 2009*) due to ozone depletion, but the latitudinal gradient of decreasing insolation from the equator to the poles is valid throughout the breeding ranges of *Spheniscus* penguins (0°–55°S), and the ozone depletion is recent and unlikely to affect foot color evolutionarily.

Another hypothesis for the function of dark pigmentation that is commonly proposed is protection from mechanical damage or abrasion (*Savalli, 1995*; *Galván & Solano, 2016*). We did not test this hypothesis because the soles (ventral surfaces) of the feet are black or

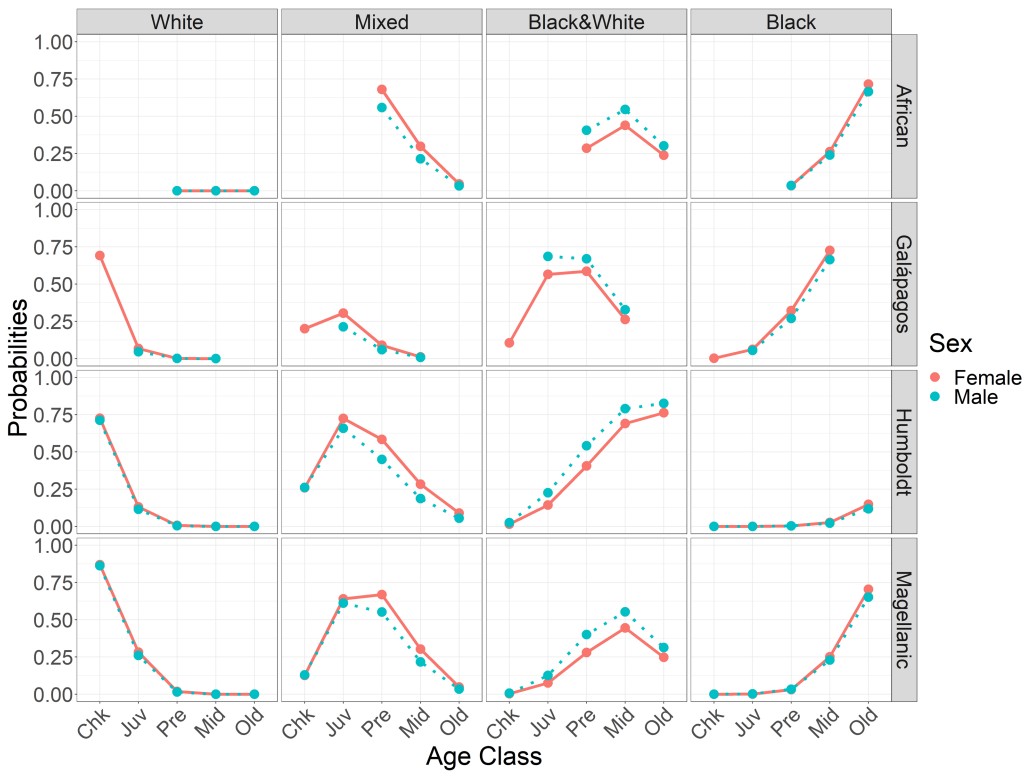

**Figure 5** **Probabilities of having each foot color in *Spheniscus* penguins, by age class and sex for each species.** Age classes: Chk, chick; Juv, juvenile; Pre, pre-breeder; Mid, middle age; Old, elder penguin; (see Table 2 for age class definitions).

dark on all penguins, even though the tops (dorsal surfaces) of the feet vary in color among species, from pink to bright orange to black (*García Borboroglu & Boersma, 2013*). The dark soles likely help protect feet from the rough surfaces penguins walk on, but the tops of the feet require less protection from abrasion. The tops of the penguin feet rarely get scraped by rocks even when they dig nests (Boersma and Rebstock pers. obs., 1983–2023). If the color of the tops of the feet protect them from abrasion, we would expect penguins that occur in rocky habitats, such as rockhopper penguins (*Eudyptes chrysocrome*), to have darker feet. However, the tops of rockhopper penguins' feet are pale (*García Borboroglu & Boersma, 2013*).

We did not consider a signaling function for foot color, as most colors used in signaling are bright, such as the feet of blue-footed boobies *Sula nebouxii* that are used in mate assessment (*Torres & Velando, 2003*). However, if foot darkness indicates age, it might be used as a signal by *Spheniscus* penguins seeking a mate of an optimal age. We think this is unlikely as foot color is a good predictor of population age structure, but not of an individual's age. A penguin using foot color to judge the age of a potential mate would frequently be wrong.

Banded penguins usually nest in burrows, crevices, or under vegetation (*García Borboroglu & Boersma, 2013*), where their feet are protected from UV radiation. Males

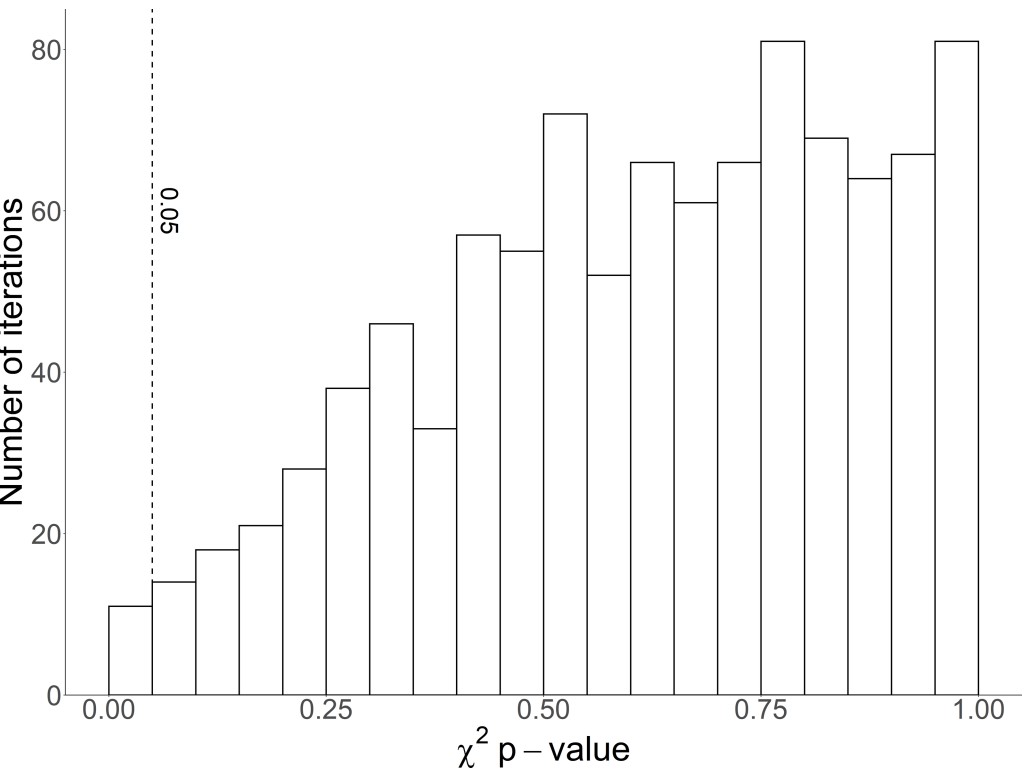

**Figure 6** **Histogram of $\chi^2$ $p$-values from 1,000 random iterations of age-structure predictions for Magellanic penguins.** The vertical dashed line marks $p = 0.05$.

spend more time on land outside their nests than females and may need more protection from UV radiation ($H_3$). In the seasonally breeding Magellanic penguin, males typically arrive at breeding colonies earlier than females (*Boersma, Stokes & Yorio, 1990*) and leave to overwinter at sea a little later (*Rebstock & Boersma, 2023*). Males also spend more time outside nests calling and fighting than females (*Renison, Boersma & Martella, 2003*; *Clark, 2006*). Males start breeding at older ages than females and are less likely to get mates (*Clark, 2006*; *Boersma et al., 2013a*). In colonies of Humboldt and Magellanic penguins with more males than females, unmated males spend more time outside of nests than mated penguins (*Taylor, Leonard & Boness, 2001*; *Gownaris, García Borboroglu & Boersma, 2020*).

Skin color changes with development in many birds, but often from darker colors in nestlings to lighter colors in adults (*Fox, 1962*; *Nicolaï et al., 2020*). Newly hatched banded penguins are protected from the sun by their parents and nests, and they have pale feet, similar to flamingos that hatch with pale legs that darken when chicks are no longer brooded (*Fox, 1962*). Magellanic penguin chicks' feet accumulate dark spots as they age.

We found that foot color darkened over years, but not within seasons ($H_4$). Skin in nonhuman animals, as well as humans, darkens over hours to days (tans) in response to UV exposure (*Lowe & Goodman-Lowe, 1996*), but we did not find evidence of seasonal or temporary darkening of penguins' feet. Seasonal changes in bill color in European starlings (*Sturnus vulgaris*) seem to be related to diet and the need for a stronger bill (more

melanin) in winter (*Iverson & Karubian, 2017*). Penguins spend more time at sea when not breeding than when they are incubating eggs and caring for chicks, so they should not need abrasion-resistant skin on their feet when not breeding.

We propose that the foot darkening is an evolutionary adaptation. Nestlings do not need pigmented feet as they are protected from UV radiation by parents and nests for much of the time before they leave the colony. Juveniles and pre-breeders spend more time at sea than breeding adults. Hence, the need for UV protection in the feet increases with age and with lower latitudes of the breeding range. Our results are consistent with this interpretation. In addition, the species that breeds at the highest latitudes, the Magellanic penguin, is more migratory (at sea) and has a shorter breeding season than the other species. UV exposure is reduced at sea as the feet are shaded by the body and penguins frequently dive below the depth that UV radiation penetrates (*Tedetti & Sempéré, 2006*) during the day. Hence, time spent at sea may combine with less UV radiation at higher latitudes to reduce the need for dark pigment in Magellanic penguins' feet. Finally, both the zoo and wild populations followed the predicted patterns of darkening. The San Diego Zoo (32.7°N) is within the (northern-hemisphere equivalent) latitudinal range of breeding African penguins in the wild. The Woodland Park Zoo (47.7°N) is at a higher latitude than Humboldt penguins' breeding range, but feet in our sample of Humboldt penguins darkened at the predicted intermediate timing. UV exposure may be the cause of the accumulation of pigments in the feet, but we propose that the timing of darkening among species is controlled by evolutionary adaptation.

Most, if not all, banded penguin chicks hatch with white feet, and the probability of white feet in chicks should be close to 1. The regression underestimated the probability because of the low number of chicks of known sex in our data. The number of chicks we observed, including those we did not use because we did not know their sex, was 10 Galápagos penguins, 16 Humboldt penguins, and 1,236 Magellanic penguins and all had white feet. Most of the Galápagos and Magellanic penguin chicks were not included in the regression (or Table 3), however, as we did not know their sexes.

We showed that age structure, but not individual age, can be estimated accurately from a sample of foot colors in Magellanic penguins. Age structure could likely be estimated in other banded penguins as well. An individual cannot be aged accurately by its foot color, similar to the case of bare-part color in other bird species (*Sellers, 2009*). Applying our findings to individual penguins and assuming that pre-breeders have mixed feet, middle-age breeders have black & white feet, and elder breeders have black feet, 30–60% of individual Magellanic penguin adults could be misclassified based on their foot color. Estimating age structure, however, does not assume that penguins in each age class have the same foot color, but uses proportions of foot color within age classes. Using proportions means that population age structure can be estimated with a high degree of certainty. Assessing foot color is a rapid, noninvasive method of estimating age structure in a population of Magellanic penguins, and likely other *Spheniscus* penguins.

## ACKNOWLEDGEMENTS

We thank the La Regina family, Esteban Frere, Popi García Borboroglu, William Conway, Carlos García, Graham Harris, Patricia Harris, Carol Passera, and Centro Nacional Patagónico (CENPAT) for logistical support and help over the years of field work in Argentina. We are grateful to the many students and field volunteers who helped collect data including students from Puerto Madryn and Trelew, Argentina. We thank Michael Pittman and two anonymous reviewers for their comments and suggestions. We thank Galo Quezada and Godfrey Merlen for help with applying for permits and making logistical arrangements in the Galapagos.

### Funding

The work in Argentina was funded by the Wildlife Conservation Society (WCS); the Benedict Family Foundation; the Disney Worldwide Conservation Fund; the Hugh and Jane Ferguson Foundation; the Offield Family Foundation; the Pincus Family Foundation; the Tortuga Foundation; Wolf Creek Charitable Foundation; the Wadsworth Endowed Chair in Conservation Science; and Friends of the Penguins. The work in the Galápagos was funded by the David and Lucile Packard Foundation, Leiden Conservation Foundation, Galápagos Conservancy, Detroit Zoological Society, and the Sacramento Zoological Society. The funders had no role in study design, data collection and analysis, decision to publish, or preparation of the manuscript.

### Grant Disclosures

The following grant information was disclosed by the authors:
Wildlife Conservation Society.
Benedict Family Foundation.
Disney Worldwide Conservation Fund.
Hugh and Jane Ferguson Foundation.
Offield Family Foundation.
Pincus Family Foundation.
Tortuga Foundation.
Wolf Creek Charitable Foundation.
Wadsworth Endowed Chair in Conservation Science.
Friends of the Penguins.
David and Lucile Packard Foundation.
Leiden Conservation Foundation.
Galápagos Conservancy.
Detroit Zoological Society.
Sacramento Zoological Society.

### Competing Interests

Ginger A. Rebstock and P. Dee Boersma are Academic Editors for PeerJ.

## Author Contributions

- Ginger A. Rebstock conceived and designed the experiments, performed the experiments, analyzed the data, prepared figures and/or tables, authored or reviewed drafts of the article, and approved the final draft.
- K. Pearl Wellington conceived and designed the experiments, performed the experiments, prepared figures and/or tables, authored or reviewed drafts of the article, and approved the final draft.
- P. Dee Boersma conceived and designed the experiments, performed the experiments, authored or reviewed drafts of the article, and approved the final draft.

## Animal Ethics

The following information was supplied relating to ethical approvals (i.e., approving body and any reference numbers):

The Institutional Animal Care and Use Committee of the University of Washington approved this research (Protocol 2213-02: Penguin Project; TR202100000016).

## Data Availability

The raw data and code are available in the Supplemental Files.

## Supplemental Information

Supplemental information for this article can be found online at http://dx.doi.org/10.7717/peerj.17937#supplemental-information.

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
