# Peer review of "Foot darkening with age in Spheniscus penguins: applications and functions"

_PeerJ, doi:10.7717/peerj.17937_

## Round 0.1 · original submission · Major Revisions

Thank you very much for your manuscript titled “Rates of foot darkening with age in banded (Spheniscus) penguins suggest that foot pigments provide UV protection” that you sent to PeerJ.
In this study, changes in foot color of the penguin Spheniscus were analyzed based on wild and captive populations of other species in North America. The results suggest that the darkening of foot occurs with age, probably due to exposure to rays UV. Foot darkening is proposed as a rapid and non-invasive way to assess age structure in ringed penguins.
As you will see below, comments from two referees suggest a minor revision before your paper can be published, but a third reviewer suggests major changes. Their comments should provide a clear idea for you to review, hopefully improving the clarity and rigor of the presentation of your work. I will be happy to accept your article pending further revisions, detailed by the referees, which largely focus on clarifying various aspects of your work.
Reviewer 1 basically suggests complementing the abstract with more detail, and some specific modifications in several lines of the manuscript.
Reviewer 2. suggests additional text and references in the introduction to expand information from existing work on foot coloration in fossil and living birds and its biological importance.
Reviewer 3 believes that part of the manuscript and some hypotheses are valid, but also believes that there are another set of topics that seem speculative and require many questionable assumptions. For example, he proposes that UV exposure might appear as a possible explanation under discussion, but not as a proven hypothesis. He also proposes detailing the abstract and modifying the title of the manuscript because he considers that "rates" were not calculated.

Please note that we consider these revisions to be important and your revised manuscript will likely need to be revised again.

Reviewer 1 ·

Basic reporting

Dear authors

Thank you for this interesting work. I've enjoyed reading it and have mostly minor comments.

The Abstract, while having all the information, could be written a bit better. I like the structure proposed here https://www.nature.com/documents/nature-summary-paragraph.pdf and suggest that the authors try and apply it here too.


Line 49: this is a repetition of line 37. Maybe start out with an even more general line 37 and then list up the specific reasons why it is important. E.g. “Knowing the age structure of a population is important for conservation as it influences multiple metrics. For example,…” Try and make the flow a bit better so that it does not feel as a list of facts.
The list of facts is a general comments and can be solved by adding small changes. E.g. by just adding “however” and “furthermore” the text reads much more pleasantly.
“Molecular techniques are increasingly being used to age wild animals (Juola et al., 2006; De Paoli-Iseppi et al., 2018). However, determining population age structure by molecular methods requires capturing large numbers of individuals and taking blood or skin samples. Furthermore, these methods do not work on all species…”
Line 91. Introduce the genus, in particular the common name (Banded penguins). It took me too long to realize that this was not about banded/ringed individuals (which of course you know the age off).
Line 101: please expand a bit why they are unlikely involved in signalling. If they change with age, can’t they signal “age” and thus fecundity ?
Line 102 extra refs.
Line 110: start with an observation, then the hypothesis that follows. (So in essence, just switch the order of the two sentences in this paragraph).
Material and methods
Please add latin names.
Line 192: out of interested. What happens with the results if you make them ordered as there is a reasonable probability that they are indeed ordered. E.g. is this not implied in line 239-240, and 247-252.
There is another hypothesis, i.e. that melanised feet heat up (either through direct sunlight, or through conductive heat transfer, see for example https://www.nature.com/articles/s42003-020-01524-w) more easily and that this helps in incubation, but I am thinking about Aptenodytes so I am not sure if this really is relevant here.

Line 325-329: This section is not completely clear.
Line 120: please clearly define age structure. I am out of this field of interest so this was sometimes confusing. Especially when it is actually discussed (line 330-340).


Again, I thank the authors for their work and wish them good luck with future research. Looking forward.

Experimental design

No comment

Validity of the findings

no comment

·

Basic reporting

Meets PeerJ standards.

Experimental design

Meets PeerJ standards.

Validity of the findings

Meets PeerJ standards.

Additional comments

This was a nice read. I especially liked how the paper provides a creative solution to broader challenges associated with studying penguin populations.

I have one suggested edit that shouldn't take long and would be really great to see. I suggest additional text and references in the introduction section to report broader existing work on foot colouration in fossil and living birds and its biological significance (Introduction paragraph on line 82 would benefit from expansion. Line 307 could be moved into this expanded intro paragraph). Suggested fossil references: Roy et al. 2020 (https://onlinelibrary.wiley.com/doi/full/10.1111/brv.12552) about fossil melanin in birds; Pittman et al. 2022 (https://www.nature.com/articles/s41467-022-35039-1) about early bird feet.

I look forward to seeing this paper in the journal soon.

Best regards,

Michael Pittman
The Chinese University of Hong Kong

Reviewer 3 ·

Basic reporting

Rebstock et al. analyzed changes in color of Spheniscus penguin feet based on two wild populations of two species, and two captive population of other two species hold in North America, to conclude that 1) darkening occurs according to age in all species; 2) UV exposure is likely the cause; 3) a population structured model is possible for Magellanic penguins. Examples of known-age penguins are used. Examples of individual birds with white feet that progressively darkened years later or from one season to the next were not provided (validation), but I assume such information in embedded in the dataset. Otherwise, we should assume that white feet penguins have very high mortality, drastically reduced from the population, what seems not reasonable. However, authors tested darkening within the same breeding season, to conclude that darkening occur along years, or, implicitly, during the non-breeding period. Authors provided robust evidences of darkening occurring through age, and provide good reasons for population structure based on such data. However, I feel that attributing to UV exposure and trying to convince readers that the latitudinal gradient (based on zoo birds), requires a range of assumptions, including duration of breeding period, lack of UV exposure when birds are at sea, among others, which seems hard to follow. In addition, there are other better ways for testing, e.g. using several colonies of a single species along wide altitudinal gradient, e.g. the Humbold penguin. Thus, in my opinion, UV exposure could appears as a potential explanation in discussion, but not as a tested hypothesis. The focus of the MS would be on demonstrating that a phenomenon occurs (in this case, feet darkening during ageing), and the applicability of such findings (in this case, population structure). Inferring that it is caused by UV exposure and the latitudinal gradient prove that, based on captivity birds in the sampling, seems far beyond evidences provided.
The MS is well presented and written, but would also benefits from using proper words in title (rate had not been calculate so far), and a result (values) in abstract, rather than convincing by arguing only. Let’ readers to be convinced by data. Other aspects which deserve attention, or additional explanations of points raised above are listed.

Experimental design

Hypothesis: a potential bias is the assumption that at low latitudes UV radiation is stronger, more intense of for longer periods, particularly due to ozone layer reduction at high latitudes of the Southern Hemisphere in recent decades. A reference or in situ measures of radiation along the latitudinal gradient would be useful to make clear that the expected latitudinal UV variation in demonstrable. An additional issue would be time of exposure of penguins. In this case, time on land could be a factor (assuming that radiation in water is reduced, not sure it is true), and species or populations breeding during shorter periods would be less affected, and so develop pigmentation more slowly. Finally, if vegetation cover or time on burrows are important at a level to influence variation in pigmentation between males and females, thus breeding duration (time exposure on land) could also be a factor. The effect of exposure to UV is not clear to me, particularly because captivity birds are involved, and the duration of the breeding period (as a proxy of exposure) is not clear. It could also differs due to migration, non-migratory species with more access to land, if this is the case (particularly for Galapagos penguins). A different interpretation would be that birds are also under strong UV radiation on water, and thus would have the top of feet less affected during most of the year (which is in opposition of some assumptions presented and would explain sole foot dark, explained in the MS as due to abrasion on land).
Study populations: two sampling places occurred in African penguins at San Diego (about 30 degrees) and Hulboldt penguins at Seattle (~45 degrees). One assumption is that UV radiation promote feed darkening, but is not clear how the use of captivity birds influenced the latitudinal gradient postulated. Furthermore, captivity birds would have distinct patterns of insolation due to artificial structures and/or time on land (much higher/throughout the year than wild birds?). How this potentially biases were addressed? Are they representing negligible influences? I would like to be convinced that findings on the latitudinal effect on pigmentation rate are still valid. This hypothesis would be better tested using a single species with larger latitudinal distribution, such as the Humboldt penguin, or even the Magellanic in several colonies.

Validity of the findings

(also included in the prvious topic, which has effects on validity)

No seasonal effect on darkening: if this is true, while more exposure account for faster darkening in males, thus exposure during breeding result in darkening in the following years, like a delayed effect of exposure… The absence of effect of exposure seems to be better demonstrable by measuring darkening degree earlier vs. later in the same season, in marked individuals. If not works as tanning in humans, and UV exposure during the non-breeding period has limited effect on darkening, darkening occurs when skin changes? During the wintering period? I’m convinced that darkening with age occurs, but the mechanism, or how it happens seems hard to follow based only on data presented.

Additional comments

Title: rate, as a calculable parameter, will indicate to readers that a value will be provided; or even better, an equation allowing calculation of approximate age. But it did not appear latter on. Thus, the title cold be more precise by avoiding the use of ‘rate’, and indicating darkening according to age classes or a similar phrasing.
Abstract: would benefit substantially of more detailed data, departing form the study design, sample sizes in each colony/place (or captivity), how many birds were followed and for how long, etc. The focus on providing findings (potentially derived from limited observations or questionable sampling, e.g. captivity birds in other latitudes), could guide readers to wrong directions.

---

## Round 0.2 · Minor Revisions

Thank you again for your manuscript titled “Foot darkening with age in Spheniscus (“banded”) penguins: applications and functions” that you sent to PeerJ.

As you will see below, reviewers 2 and 3 consider some minor corrections necessary for the final acceptance of the manuscript.

I ask you to please address these comments so I can accept his article.

·

Basic reporting

No comment

Experimental design

No comment

Validity of the findings

No comment

Additional comments

The authors added a phrase and accompanying suggested reference in response to my review. The authors also made other changes in response to Reviewers 1 and 3. The manuscript has improved. I recommend acceptance and congratulate the authors on their work. I look forward to seeing it in print.

Reviewer 3 ·

Basic reporting

As a second round review I found explanations in the reply letter and further, much necessary, clarification in the main text, satisfactory.

Further minor suggestions are:

Title: Despite ‘banded’ requires clarification, it would be done later in introduction, not in title. ‘Spheniscus penguins’ sounds better.

Abstract: L. 25 (lines as in the tracked changes version). Add latin names at first mention.
Throughout the abstract and the MS, common names are both capitalized and non-capitalized (e.g. L. 31 vs. 33; L. 678 vs. 679). Use the standard requested by the journal, or at least be consistent throughout.

Discussion: L. 422-426. I agree with this interpretation in penguins, but other species, e.g. blue-footed boobies signal partners using feet colour, as vastly demonstrated by Drummond, Velando, Torres and other researchers. I would suggest make clear that this is interpretation is valid for Spheniscus penguins, but differ from other seabirds.

Experimental design

NA

Validity of the findings

NA

Additional comments

NA

---

## Round 0.3 · accepted · Accept

After checking the latest corrections made to your manuscript, I am pleased to see that it now meets the conditions for publication. Congratulations.